# Combining Electrostatic, Hindrance and Diffusive Effects for Predicting Particle Transport and Separation Efficiency in Deterministic Lateral Displacement Microfluidic Devices

**DOI:** 10.3390/bios10090126

**Published:** 2020-09-16

**Authors:** Valentina Biagioni, Giulia Balestrieri, Alessandra Adrover, Stefano Cerbelli

**Affiliations:** Dipartimento di Ingegneria Chimica Materiali Ambiente, Sapienza Università di Roma Via Eudossiana 18, 00184 Roma, Italy; valentina.biagioni@uniroma1.it (V.B.); balestrieri.1653798@studenti.uniroma1.it (G.B.); alessandra.adrover@uniroma1.it (A.A.)

**Keywords:** cell sorting, label-free, fractionation, size-based, convection-enhanced dispersion

## Abstract

Microfluidic separators based on Deterministic Lateral Displacement (DLD) constitute a promising technique for the label-free detection and separation of mesoscopic objects of biological interest, ranging from cells to exosomes. Owing to the simultaneous presence of different forces contributing to particle motion, a feasible theoretical approach for interpreting and anticipating the performance of DLD devices is yet to be developed. By combining the results of a recent study on electrostatic effects in DLD devices with an advection–diffusion model previously developed by our group, we here propose a fully predictive approach (i.e., ideally devoid of adjustable parameters) that includes the main physically relevant effects governing particle transport on the one hand, and that is amenable to numerical treatment at affordable computational expenses on the other. The approach proposed, based on ensemble statistics of stochastic particle trajectories, is validated by comparing/contrasting model predictions to available experimental data encompassing different particle dimensions. The comparison suggests that at low/moderate values of the flowrate the approach can yield an accurate prediction of the separation performance, thus making it a promising tool for designing device geometries and operating conditions in nanoscale applications of the DLD technique.

## 1. Introduction

An ever growing number of analytical essays and preparative processes in clinical and biological practice hinge on the label-free, size-based separation of mesoscopic objects suspended in a buffer solution. Classical and widely used methods to carry out these types of separations are microcentrifugation [1], ultrafiltration [2], and SEC (Size Exclusion Chromatography), where the solution suspending the particles is pushed through a column packed with porous grains [3]. However, well characterized and standardized for a wealth of target applications, each of these techniques is afflicted by specific shortcomings. Ultrafiltration is a two-step operation in that the targets that are retained at the membrane must be released to be recovered. Microcentrifugation can be very effective in separating a size-dispersed suspension, yet is time-consuming and typically limited in throughput. As regards SEC-based separations, their major shortcoming is the giant axial dispersion coefficient that characterizes the transport of the suspended particles that fall below the cutoff length for entering the grain pores. If one considers one such particle suspended in the mobile phase near the pore entrance of a generic grain, it is evident that a small fluctuation can cause the either the particle to enter the grain or to continue its route with the mobile phase, thus creating bifurcating paths that are associated with altogether different residence times within the packed bed. By this mechanism, the axial dispersion coefficient characterizing particle transport in the disordered structure can result in being orders of magnitude larger than the bare diffusion coefficient of the particle. Clearly, the amplification of axial dispersion hinders separation resolution and boosts the HETP (Height Equivalent Theoretical Plate) of the column [3].

In the last few decades, the development of computer-assisted microfabrication techniques has made it possible to conceive alternative solutions to the size-based separation of suspensions and colloids. Among the methods proposed, hydrodynamic chromatography (HDC) [4,5], and, more recently, Deterministic Lateral Displacement (DLD) [6], have motivated an intensive research interest. In both cases, the effect of downsizing the separation equipment to the microfluidic domain allows for gaining precise control over the flow, which is inherently laminar at these lengthscales. In this regime, the flow structure can be either predicted analytically, or else can be obtained through the numerical solution of the Stokes equations at modest computational effort, even when complex device geometries are being dealth with.

Analytical solutions can readily be obtained for HDC columns, which are simple open capillaries hosting unidirectional Poiseuille flow of the suspending solution. The separation mechanism is here based on the fact that the larger particles cannot experience the near wall low-velocity region as they diffuse accross the channel section because of the hindrance effect due to their size [4]. Thus, large particles possess an average velocity higher than solvent molecules (the larger, the higher) and yield a chromatographic response where the average particle residence time continuously depends on particle size. Even though convection-enhanced dispersion occurs in these types of devices (the classical Taylor–Aris dispersion phenomenon), the amplification of the axial dispersion coefficient is here less severe with respect to SEC separations (see, e.g., [7] and therein cited references), especially when finite-sized particles are being processed [8]. In addition, SEC-based separations still rest upon a transient regime, and, since the average particle velocity depends weakly on the particle size, they require long channels to achieve a practically relevant separation resolution.

All of these shortcomings are bypassed in DLD devices, where the size-based separation occurs continuously in time, typically under steady-state conditions (see [9,10] for an updated account of of the state of the art on the subject). As far as applications are concerned, DLD-based separations have been implemented for sorting, concentrating, and isolating a wide range of analytes of clinical/biological interest that behave as suspended particles, such as leukocytes [11], cancer cells [12,13,14,15], DNA fragments [16], exosomes [17], fungal spores [18], parasites in human blood [19], subpopulations of large-sized cells to be used in tissue engineering [20], and blood cell types from whole blood [21,22].

The core of the device is a shallow channel with a rectangular cross-section filled with a spatially periodic array of impermeable obstacles arranged according to a two-dimensional Cartesian lattice, not necessarily orthogonal. The obstacles are typically cylindrical, even though different shapes have been considered for reducing the pressure-drop and preventing possible clogging of the interstitial channels by particle clustering [23,24].

One of the axes of the Cartesian array is slanted by an angle $\Theta_{l}$ with respect to the average flow direction which is constrained by the lateral walls of the channel (Figure 1a). The separation principle is based on the bifurcating structure of the pressure-driven laminar flow through the array, on the one hand, and on the particle–obstacle interactions, on the other. Specifically, assuming that the tilting angle is defined by the ratio of two integers, $\tan(\Theta_{L})=M/N$ (M<N), one can identify *N* different fluxtubes (referred to as *lanes*), delimited by critical streamlines that repeat themselves periodically throughout the lattice, as depicted in Figure 1b for the case M=1, N=4. In the large (i.e., at the scale of the device, which hosts order $10^{5}$ obstacles), each lane runs parallel to the side walls of the channel. In a highly idealized picture of particle motion, referred to as the *kinematic approach*, the center of a (supposedly spherical) particle is assumed to track the streamlines of the unperturbed single phase Stokes flow. This is so long as the particle surface does not come into contact with the obstacle surface. On surface contact, the particle glides around the obstacle under the simultaneous action of viscous drag and of particle–obstacle interactions, and eventually detaches from it until a collision with a different obstacle occurs downstream the array. A representation of the behavior predicted by this approach for particles of different size is depicted in Figure 1c,d.

When colliding with the obstacle, the smaller particle deviates from the behavior of a massless tracer ideally located at its center, which would be constrained to track a fixed streamline of the flow. However, this deviation is not enough to make the center of the particle cross the critical streamline and change lanes. Thus, a particle of this size never changes its fluxtube, and, likewise, the lane hosting the particle trajectory, its average direction downstream the device, is collinear to the side wall of the channel. The situation is altogether different for a large particle. Here, hindrance effects force the center of the particle to change fluxtubes at each and any obstacle collision (Figure 1d). As a consequence, the average direction of the particle at a large scale will be parallel to the axis of the Cartesian obstacle lattice and hence will deviate by an angle $\Theta_{L}$ with respect to the average flow direction. In this simplified kinematic picture, any obstacle lattice is associated with a critical particle size, say $R_{\mathrm{crit}}$, which is given by the thickness of the fluxtube next to the obstacle (see Figure 1b). Based on this microscale kinematic model, the separation process can be run continuously in time by simply introducing a focused stream of the buffer suspending a mixture of particles of different size upstream the obstacle lattice, and collecting particles of subcritical and supercritical size at different cross-sectional locations at the device outlet, which are defined by the lattice angle $\Theta_{L}$ as depicted in Figure 2.

Note that, within this simplified modeling approach, all particles whose radius is smaller than the critical radius $R_{c}$ travel along a single trajectory characterized by zero deflection angle, whereas particles of radius larger than $R_{c}$ travel along a single trajectory characterized by a deflection angle $\Theta_{L}$. It is also worth observing that, so long as the flow is in the creeping flow regime, this feature is independent of the flowrate of the buffer solution. The left panel of Figure 2 depicts the qualitative features that are typically observed in the real world separation process. Here, particles of each given size scatter about a median trajectory, whose direction continuously depends on the specific particle size. A detailed account of the phenomenology of particle transport is described in the next section. Here, we just highlight that this phenomenology must be accounted for in theoretical models of particle dynamics that are meant to provide a design framework for specific applications of the DLD separation technique.

In Section 2, we build up from existing models of particle motion in DLD devices, each focusing on a specific aspect of the dynamics, and construct a template that can quantitatively predict the performance of these microseparators. The fitness and flexibility of the approach proposed are tested against experimental data in Section 3, by using the results reported in [6] as a benchmark. We show that our approach improves by far the prediction of any previous model proposed in the literature, including those previously developed by our group.

## 2. Phenomenological Aspects of Particle Dynamics and the State of the Art of Modeling Approaches

Before providing a concise overview of the different aspects of particle dynamics that have been modeled in the DLD literature, we briefly discuss the phenomenology observed in real world equipment by using the results of the experiments performed on Huang’s prototype [6]. In Section 4, we use these data to assess the validity of the transport model described in the next section.

### 2.1. Phenomenological Aspects of Particle Dynamics

The device geometry used in [6] is depicted in Figure 3a.

The tilting angle was here was set equal to $\Theta_{L}=\tan^{-1}\left(1/10\right)$, which gives rise to a Stokes flow characterized by ten adjacent lanes in the elementary cell (details are given in Section 2.2 below). In [6] fluorescent particles of different diameters ($d_{p}=0.6;\;0.8;\;1.0\;\mu m$ were continuously introduced in a narrow focused stream (ideally of vanishing thickness) upstream the obstacle lattice. Long-Time exposure micrographs were taken at the device cross-section 11mm downstream the entrance. From these images, the brightness as a function of the cross-sectional coordinate, *x*, was measured (Figure 3b). At steady-state, the fluorescence intensity signal at a given position between *x* and x+Δx is thus proportional to the number of particles that exit the device at this interval per unit time. (c) and (d) of Figure 3 depict the brightness measurements for a flowrate equal to 40μm/s and 400μm/s, respectively. The data clearly show that, at both flowrates, the suspension of particle is well separated at the device exit. In addition, the following observations can be made based on these data:Even if the inlet stream is focused (i.e., ideally characterized by vanishing thickness) particles of one and the same size tend to scatter transversally to the flow direction as they cross the device, so that they exit the device exit at different positions along the outlet section.The average particle deflection (i.e., the median line of each of the particle streams qualitatively depicted in Figure 2 which determines the position of the peak of the distribution at the exit cross-section) depends continuously on particle size, and varies as the flowrate of the suspending solution is varied for one and the same particle size (compare (c) and (d)).The dispersion bandwidth about the peak of the exit distribution associated with different particle sizes, which ultimately results from particle diffusion, depends on both the particle size and the flowrate of the suspending solution.The comparison between (c) and (d) of Figure 3 shows that the width of the distributions for each particle size is either unaltered or decreases very modestly when the flow rate undergoes a tenfold increase.This phenomenon unambiguously indicates the presence of a synergistic effect between the deterministic structure of the flow advecting the particle at the scale of the elementary cell and its isotropic Brownian motion component.In fact, in the case where no interaction is assumed between the two transport mechanism (and therefore diffusion could be plainly superimposed to the average particle motion), elementary arguments show that the variance of the distribution should be in the ratio $1/\sqrt{10}$ for all particle sizes, which is clearly not the case of the data shown.

In conclusion, the data show that the impact of diffusion on particle dynamics in DLD lattices are both important and nontrivial, thus suggesting that this aspect of particle transport should not be overlooked when designing the lattice geometry and fixing the operating conditions of the microseparator.

### 2.2. Overdamped (Kinematic) Motion

Given the above picture, it is worth comparing and contrasting the experimental results with the prediction of the kinematic approach for the system at hand. (a) and (b) of Figure 4 show the structure of selected critical streamlines (i.e., of the streamlines that originate from, and terminate onto, a solid obstacle) for the Stokes flow through the obstacle lattice defining Huang’s device. Streamlines repeat themselves every tenth row of obstacles downstream the flow (here supposed directed from vertically from the top to the bottom of the picture). (b) depicts the lanes defined by the critical streamlines originated upstream the restricted gap.

Because the lattice angle satisfies $\tan(\Theta_{L})=1/10$, the flow through the elementary cell can be broken down into ten fluxtubes, each carrying one tenth of the total flowrate through the cell. If one assumes that the inlet stream suspending the particles is ideally of zero thickness, then all of the particles can be regarded as entering (one at a time) the lattice at x=0. Owing to the strictly deterministic character of the transport model, particles that are below the critical radius will all exit the device at x=0, whereas particles that are above the critical size should all exit the device at $x_{L}=L\tan\left(\Theta_{L}\right)=1/10=1100\;\mu m$ (*L* is the distance between the feeding point and the lattice outlet in the streamwise direction, see Figure 3b). Therefore, the kinematic model applied to Huang’s experiments predicts that the (normalized) exit particle distribution, say $P(x;r_{p})$ ($r_{p}$ being the particle radius) is given by a Dirac delta function centered either at x=0 (for subcritical particles), or at $x_{L}$ (for particle radii larger than the critical value)
(1)$$P\left(x;r_{p}\right)=\left\{\begin{array}{ccc} \delta (x) & \mathrm{for} & r_{p}<R_{c}\\ \delta \left(x-L\;\tan(\Theta_{L})\right) & \mathrm{for} & r_{p}>R_{c} \end{array}\right.$$

It is worth highlighting that the flowrate does not appear in Equation (Equation 1). This can be readily understood by considering that, in the Stokes regime, increasing the flow rate increases the velocity magnitude while leaving the flow geometry unaltered. Because in the kinematic approach the fate of a particle of a given size is only determined by the obstacle geometry and the streamline structure of the unperturbed Stokes flow, the velocity magnitude cannot induce any effect of the particle path. Thus, however important for highlighting the fundamental mechanisms driving the size-based separation of particles, the kinematic approach hardly provides a useful quantitative tool for the actual design of DLD devices.

### 2.3. Diffusion-Induced Effects

The phenomenological behavior of particles observed in the experiment described above unambiguously indicates that particle diffusion cannot be neglected when building a realistic transport model. Generally speaking, the analysis of the interaction between advection and diffusion in spatially-periodic flows is a subject of paramount importance in transport theory since it constitutes the simplest possible framework to approach a variety of situations that involve porous media, from the analysis of transport-controlled catalytic reactions, to dispersion of contaminants in ground soil. The behavior of massless (i.e., point-sized) particles, also referred to as point *tracers*, has been extensively investigated in the last four decades [25,26]. Anisotropic enhanced dispersion regimes, where the entries of the dispersion tensor can be orders of magnitude higher than the bare particle diffusivity, have been identified and predicted using different approaches. The case of finite-sized particles, which is directly related to the performance of DLD devices, has received comparatively less attention, the first works on the subject being focused only on determining the impact of particle diffusion on the average deflection angle of the particle current for each given particle size [27]. In a series of previous articles [28,29,30], our group developed a complete quantitative framework to estimate the fundamental transport parameters that control particle dynamics at the device scale (i.e., at lengthscales much larger then the size of the unit fundamental cell of the lattice). These parameters consist of the average particle velocity (a vector quantity) and the particle dispersion tensor, a tensor quantity that accounts for the non-isotropic dispersion behavior observed at large scales.. We showed how these quantities depend sensitively on particle size, on the average flowrate and on the lattice geometry. In [30], we also discussed how the average transport parameters can be used to interpret, characterize, and predict the exit distribution of the particles in a typical DLD separation operation. To develop this transport template, we used a microdynamical model for particle evolution where the particle–obstacle interaction is modeled as a hard-wall potential: the particle follows passively the unperturbed flow (computed at the particle center) until it collides with an obstacle. Here, only the component of the flow orthogonal to the touching surfaces is conserved, whereas the component normal to the surfaces is annihilated. Thus, in the absence of diffusional motion components, the model provides the quantitative counterpart of the qualitative kinematic picture based on the concept of lanes that was first proposed in [6]. However, Brownian particle motion can be readily introduced in the physical framework as a stochastic term to be added to the deterministic particle velocity arising form the fluid drag and the particle–obstacle interaction:(2)$$dx_{p}=v\left(x_{p}\right)\;dt+\sqrt{2\;D_{p}}\;d\xi$$

In Equation (Equation 2), $x_{p}$ denotes the (x,y) coordinate position of the center of the particle, $D_{p}$ is the particle bare diffusion coefficient, and dξ is the increment of the vector-valued Wiener process in the time interval dt, possessing zero mean and variance equal to $\sqrt{dt}$, which models the effect of particle Brownian motion (for alternative approaches, see [31] and therein cited references)

Note the particle size does not enter explicitly Equation (Equation 2) in that the equation defines the dynamics of a point, but only indirectly through the notion of *effective obstacle*. Assuming that the particle is spherical in shape with radius $r_{p}$, the effective obstacle is the surface obtained by displacing each point of the (physical) obstacle surface by a distance equal to $r_{p}$ in the direction locally normal to the surface itself. Thus, if obstacles are cylindrical in shape with radius $R_{o}$, then the effective obstacles as seen by a particle of radius $r_{p}$ are cylinders of radius $R_{o}+r_{p}$. Because the center of the particle cannot cross the effective obstacle boundary, an elastic reflection at this boundary in enforced when advancing the center of the particle through Equation (Equation 2).

Owing to the stochastic character of Equation (Equation 2), physically meaningful features of particle dynamics can be defined only in a statistical sense, i.e., by considering averages over a large number of independent realizations of the stochastic process. Figure 5 shows the instantaneous positions of particles of two different sizes (red $r_{p}=0.45\;\mu m$; blue $r_{p}=0.30\;\mu m$) that were released at the origin of the coordinate system for a statistically significant number of realizations (order $10^{5}$) of the stochastic process in Huang’s device [6] for a flowrate of 40μm/s.

Snapshots of the particle swarms are depicted at three time instants. At the (device) scale of the figure, the particle ensembles swiftly attain the form of elongated ellipses which translate at constant velocity, say $W=(W_{x},W_{y})$. The axes of the ellipses grow as $\sqrt{2\;d\;t}$ and $\sqrt{2\;D\;t}$, where d,D are the eigenvalues of the effective dispersion tensor (d≤D). Thus, the dynamics of the ensemble at these lengthscales is governed by an effective advection–diffusion process characterized by a constant average particle velocity and an anisotropic dispersion. From the motion of the center of mass of the swarm and of the variances, the average particle velocity and the dispersion coefficient can be computed through a best fit of the linear scalings of these quantities (see Section 4 for details). In turn, these transport parameters can be exploited to predict the steady state distribution along the device cross-section at assigned distance *L* downstream the injection point, which ideally attains the shape of a Gaussian distribution, centered at ${\bar{x}}_{L}=\left(W_{y}/W_{x}\right)\;L$ with variance ${\bar{\sigma }}_{L}$,
(3)$$P\left(x\right)=\frac{1}{\sigma_{L}\sqrt{2\;\pi }}\exp\left(-\frac{1}{2}{\left(\frac{x-{\bar{x}}_{L}}{\sigma_{L}}\right)}^{2}\;\right)$$

In Equation (Equation 3), $\sigma_{L}=\sqrt{2\;D_{\mathrm{eff}}\;L}$, where the effective steady-state dispersion coefficient $D_{\mathrm{eff}}$ depends on the entries of the dispersion tensor and on the direction of the average particle velocity [30]. In the case where the lattice angle $\Theta_{l}$ is small, $D_{\mathrm{eff}}$ can be approximated by the dispersion coefficient $D_{xx}$ of the swarm along the direction orthogonal to the average flow direction, which can be computed based on the evolution of the ensembles depicted in Figure 5.

In previous work by some of these authors, it has been shown that the transport template described above, which explicitly accounts for the finite size of the particle and for its diffusional motion components, provides a model that qualitatively predicts all of the transport effects described in points 1–6 of Section 2.1, namely the dependence of the average particle migration angle on both particle size and flowrate, as well as the occurrence of enhanced dispersion regimes resulting from the interaction between deterministic motion and Brownian fluctuations at lengthscales comparable to the size of the fundamental periodic cell. In addition, this template is completely predictive in that it does not involve the estimate of adjusting parameters: in point of fact, the only pieces of information needed are the single-phase laminar flow through the periodic cell (which, owing to the linear character of the Stokes problem, can be computed with accuracy at modest computational effort), the particle size and its bare diffusion coefficient $D_{p}$, which can be estimated from the Stokes–Einstein relation, $D_{p}=k_{B}\;T/\left(6\pi r_{p}\mu \right)$ (here $k_{B}$ and μ represent the Boltzmann constant and the dynamic viscosity of the fluid, respectively).

However, qualitatively consistent with the physical phenomenology, predictions stemming from the advection–diffusion model described above are not quantitatively consistent with experimental results, especially as regards the dependence of the average deflection angle on particle size and flowrate. Reasons for the quantitative discrepancy between measured and predicted values can be ascribed to different physical effects that are not modeled by the advection–diffusion model, namely

Two-phase effects, which alter the streamline geometry of the flow and make particle dynamics nontrivial. Such effects are expected to be important for large particles and high flowrates.Particle inertia, which makes the particle deviate from the underlying structure of the unperturbed flow. This effect is expected to be significant only at relatively large values of the Reynolds number, a situation that is quite uncommon in practical implementations of the DLD technique.Surface interactions, which may have an impact on the minimal distance between the particle and the obstacle surface. Note that, in the advection–diffusion model described so far, this distance vanishes during the collision.

Including full two-phase coupling and inertia in particle dynamics is a non-trivial task, both because of numerical complexity and of the increased number of parameters entering the dynamical model. In addition, the more recent trends of DLD separations are shifting the domain of application of this technique from micrometer- (cells, parasites, bacteria) to nanometer-sized objects (viruses, exosomes, RNA, DNA). In this range of lengthscales, it is sensible to assume that inertial and two-phase effect might be secondary, at least so long as the target objects can be assumed non-deformable (see, e.g., [32] for the assessment of the coupling between particle-shape and flow of the suspending solution). By contrast, electrostatic effects, which depend on surface charges, may become determinant in that the volume-to-surface ratio increases when downscaling the device. Next, we discuss the results of recent studies investigating electrostatic effects in DLD devices. These results will be used in Section 3 for constructing a particle transport template that overcomes the limitations of previous models and is able to provide a quantitative description of the particle transport phenomena observed in experiments at low flowrates.

### 2.4. Electrostatic Forces

Amongst the fluid-mediated surface interactions between solids, electrostatic forces are expected to be of primary importance. In the specific context of DLD devices, recent articles by Zeming et al. [33,34] showed that particle–obstacle electrostatic interactions can significantly impact the dynamics of suspended particles in DLD arrays. These interactions originate from the existence of an Electric Double Layer (EDL) at the solid–fluid interface. The overall effect of the EDL is to induce a net charge distributed onto the solid surfaces, characterized by a surface charge density $\rho_{\mathrm{el}}$. In the case where the sign of $\rho_{\mathrm{el}}$ is the same for both the obstacle and the particle surfaces, a repulsive force arises, which prevents the particle to collide with the obstacle. In [33], the authors proposed a simple lumped approach to account for electrostatic forces based on the concept of electrostatic displacement $d_{F-\mathrm{ELD}}$. This quantity is defined as the equilibrium distance between the obstacle boundary and the surface of a particle, which is subject to the viscous drag force of the surrounding fluid flowing with characteristic velocity *V* on the one hand, and of the electrostatic repulsion between the charged surfaces on the other, as depicted in Figure 6.

It can be shown that the electrostatic displacement does not depend on the particle radius and can be computed as:(4)$$d_{F-\mathrm{ELD}}=-\lambda_{D}\log\left(\frac{-\rho_{\mathrm{el},p}^{2}\;\rho_{\mathrm{el},s}^{2}+\sqrt{\rho_{\mathrm{el},p}^{4}\;\rho_{\mathrm{el},s}^{4}+\frac{\left(\rho_{\mathrm{el},p}^{2}+\rho_{\mathrm{el},s}^{2}\right)\;3\;\mu \;\varepsilon_{r}\;\varepsilon_{o}\;V}{\lambda_{D}}}}{\rho_{\mathrm{el},p}^{2}+\rho_{\mathrm{el},s}^{2}}\right)$$

Here, $\rho_{\mathrm{el},p}$ and $\rho_{\mathrm{el},s}$ are the surface charge density of the particle and of the obstacles surfaces, respectively, $\lambda_{D}$ is the Debye length (which depends on the ionic strength of the solution), $\varepsilon_{r}$ is the dielectric permittivity of the solution, and $\varepsilon_{o}$ is the permittivity of free space. In the presence of equally charged surfaces (i.e., positively or negatively), the electrostatic displacement can be considered as a hard core shell surrounding the particle, which is therefore characterized by an effective radius, say $r_{p}^{\mathrm{eff}}$, given by
(5)$$r_{p}^{\mathrm{eff}}=r_{p}+d_{F-\mathrm{ELD}}$$

## 3. Transport Model and Exit Particle Distributions

Next, we combine the lumped approach by Zheming et al. to model electrostatic interactions with the advection–diffusion model based on the concept of effective obstacle developed in previous articles by our group.

The effective obstacle is here defined based on the effective particle radius given by Equation (Equation 5). For fixed lattice geometry, particle size, the nature of the suspending solution and flowrate, the prediction of the particle exit distribution is based on the following steps:Compute the single-phase Stokes flow within the unit periodic cell with no slip boundary conditions on the boundary of the physical obstacle and with periodic boundary conditions for the velocity components enforced between the opposite edges of the cell. Because of the presence of the lateral channel walls defining the direction of the average flow, a global constraint on the direction of the average pressure drop must also be enforced (e.g., with reference to Figure 3, and the average pressure drop must be directed along the *y*-axis).Based on the nature of the solid surfaces and the ionic concentration of the buffer solution suspending the particle, compute the surface densities $\rho_{\mathrm{el},p}$ and $\rho_{\mathrm{el},s}$, and the bare particle diffusivity from the Stokes–Einstein relation. Then, use Equation (Equation 4) to compute the electrostatic displacement, $d_{F-\mathrm{ELD}}$ and obtain the effective particle radius defined by Equation (Equation 5). For a particle of assigned size, the effective particle radius defines the extent of the effective obstacle depicted in Figure 7 and therefore the region accessible to the center of the particle.Initialize the position of the center of the particle at the release point (here the origin of the coordinate system), and use Equation (Equation 2) to advance this position in time for a (small) finite interval Δt. Because the solution of the velocity field v is obtained numerically, an interpolation scheme is necessary for computing $v(x_{p})$ at a generic position $x_{p}$.If the advanced position falls beyond the boundary of the effective obstacle, reflect the portion of the displacement vector that falls within the boundary about the local normal vector.Repeat the integration for a large number of independent realizations, say $N_{s}$, of the stochastic process to obtain the evolution of the ensemble of trajectories as those depicted in Figure 5.Define the ensemble-average operator
(6)$$E\left[s\right]\left(t\right)=\left(1/n_{\xi }\right)\sum_{n_{\xi }}s_{\xi }\left(t\right)$$
of an observable *s* evolving alongside a generic realization ξ of the stochastic process. Track the first and second moments of the ensemble, E[x](t), E[y](t), and $E\left[x^{2}\right]\left(t\right)$. After a short transient, the first moments and the squared variance of the ensemble display a linear time scaling,
(7)$$x_{c}\left(t\right)=E\left[x\right]∼W_{x}\;t\;y_{c}\left(t\right)=E\left[y\right]∼W_{y}\;t$$
(8)$$\sigma_{x}^{2}\left(t\right)=E\left[x^{2}\right]-E^{2}\left[x\right]∼2\;D_{xx}\;t$$Here, $x_{c}$ and $y_{c}$ physically represent the center of mass of the ensemble. Thus, by tracking $x_{c}\left(t\right)$, $y_{c}\left(t\right)$, $\sigma_{x}^{2}\left(t\right)$ the the effective velocity $\left(W_{x},W_{y}\right)$ and the entry $D_{xx}$ of the (symmetric) effective dispersion tensor can be computed from the ensemble dynamics by linear regression.From the effective transport coefficients, the peak position, ${\bar{x}}_{L}$ and the variance $\sigma_{L}$ entering the steady-state Gaussian distribution in Equation (Equation 3) can be computed as
(9)$${\bar{x}}_{L}=\frac{W_{y}}{W_{x}}L$$
and
(10)$${\bar{x}}_{L}=\sqrt{2\;D_{xx}L}$$We note that the expression of the variance given by Equation (Equation 10) constitutes an approximation in that the continuous effective dispersion coefficient $D_{\mathrm{eff}}$ as defined in [30] should appear on the right-hand side in place of $D_{xx}$. However, in all of the cases where the lattice angle is small $\tan(\Theta_{L})<<1$ (which is largely verified in practical implementations of the separation method), one finds that $D_{\mathrm{eff}}≃D_{xx}$, so that Equation (Equation 10) holds true.

The transport template described above constitutes a fully predictive tool for characterizing the steady-state particle distribution of a suspension of particles of different size at the outlet of a DLD device. In the next section, we compare model predictions with the experimental results discussed in Section 2.1 to assess the validity of the approach to describe the salient features of particle motions. Because two-phase effects are not modeled, we focus on the separation performance at low flowrate, V=40μm/s, where these effects are expected to be less important.

## 4. Case Study

Figure 8b shows the solution of the Stokes problem for the pressure-driven flow through the unit cell with boundary conditions defined in the previous section, obtained by using a commercial finite element solver (Comsol Multiphysics 5.2). The unstructured mesh used for the numerical solution is depicted in (a). As can be gathered, due to the linearity of the laminar flow problem, no sharp boundary layers are observed, making it numerically straightforward to obtain the flow solution at modest computational expenses.

The velocity field thus obtained was bilinearly interpolated onto a 1000×1000 uniform square grid covering the unit cell, and continued periodically onto the lattice to obtain the velocity at an arbitrary point occupied by the center of the particle alongside the integration of Equation (Equation 2). The integration time-step was chosen so that ordering a generic particle took order $10^{3}$ time steps to cross a single unit periodic cell. Particle ensembles consisting of order $10^{5}$ independent realizations of the stochastic process were computed and recorded at regular time intervals for a flowrate V=40μm/s and values of the particle radius equal to $r_{p}=0.3;0.35;0.4;0.45;0.5\;\mu m$ to be compared to the experiments reported in [6]. In performing the integration, the estimate of the electrostatic displacement $d_{F-\mathrm{EDL}}$ was necessary in order to compute the effective particle radius which defines the size of the effective obstacle as defined by Equation (Equation 5). As can be gathered by Equation (Equation 4), $d_{F-\mathrm{EDL}}$ depends on the charge densities onto the particle and obstacle surfaces, which, in turn, depend on the nature and composition of the buffer solution suspending the particles as well as on the nature of the solids wetted by the buffer. Lacking specific information to estimate the surface densities in the experimental conditions used in [6], we proceeded as follows. We picked a specific particle size, namely $r_{p}=0.3\;\mu m$, and we tuned up the thickness of the electrostatic displacement so that the average exit position at a distance *L* downstream the feeding point obtained through the stochastic Lagrangian model would match the value determined experimentally given by the mean value of the fluorescence intensity signal. With $d_{F-\mathrm{EDL}}$ thus computed, we proceeded following steps 3 to 5 of the previous section in order to determine the steady-state particle distribution at the exit cross-section. Figure 9 shows the comparison between the experimentally determined distribution taken from [6] and that predicted by the advection–diffusion model Equation (Equation 2) solved with an effective obstacle radius taking into account the electrostatic displacement (continuous line).

Because $d_{F-\mathrm{EDL}}$ was fixed so that the average of the two distributions are equal to each other, the position of peaks nearly coincide (the small discrepancy is due to the asymmetry of the experimentally determined distribution). In addition, the good agreement between the width of the distributions is a first clear indication of the fitness of the model to describe the essential features of particle transport in DLD devices in that the variance constitutes a free parameter of Equation (Equation 3). It is worth mentioning that the estimate of the variance based on the bare particle diffusivity in place of the effective dispersion coefficient would yield a much more peaked distribution, with a variance about three times smaller with respect to the experimental value. The broken line in the same figure represents the prediction of the advection–diffusion model when the electrostatic force is not considered. As can be gathered from the comparison with the experimental data, the mean exit position is significantly underestimated when the electrostatic effect is not accounted for.

Hinging on the fact that the value of $d_{F-\mathrm{EDL}}$ is unaffected by particle size, we attempted a full prediction of the exit distributions for different particle sizes, namely $r_{p}=0.35;0.4;0.45;0.5\;\mu m$ based on the value of electrostatic displacement computed for the $r_{p}=0.3\;\mu m$ particle.

Figure 10 shows the predicted values for $r_{p}=0.35\;\mu m$ (**a**), and $r_{p}=0.4\;\mu m$ (**b**), and $r_{p}=0.45;0.5\;\mu m$ (**c**) compared to the experimentally determined distributions. The prediction for the mean exit position is excellent, whereas the values of the width of the distribution appear slightly overestimated as the particle size increases.

A more concise representation of the capability of the advection–diffusion model with an ED effect to capture the relevant features of particle transport and separation efficiency is depicted in Figure 11, which shows the mean, ${\bar{x}}_{l}$, and variance, $\sigma_{L}$ of the exit distribution as predicted by the kinematic (dash-dotted line with empty squares), the advection–diffusion model with no ED effect (dotted line with empty triangles), and the advection–diffusion model including the ED effect (broken line with empty circles) vs. the experimental values (continuous line with solid symbols).

The data make it readily evident how the advection–diffusion model with ED effect constitutes a significant improvement over all previous approaches, especially as regards the predicted mean value, for which the error is within few percent for all particle sizes. Although an appreciable overestimation of the variance for the larger particle sizes can be noted, the relative error between experimental and predicted value is still below 50%, even in the most critical cases of large size particles.

It must be mentioned that the prediction of high flowrate conditions considered in [6] (V=400μm/s) did not compare as well as above with the experimental data, as can be gathered from Figure 12.

Here, the application of the approach described above to the $r_{p}=0.3\;\mu m$ particle yielded a vanishing value for the electrostatic displacement, thus suggesting that the electrostatic forces should become unimportant at this flowrate. However, the intermediate particle sizes are significantly overpredicted. This occurrence may be attributed to the onset of different physical phenomena which may become important as the flowrate is increased and that are not accounted for in the approach proposed. Among these phenomena, we here single out two effects, one associated with the two-phase nature of the particle transport process, the other to its intrinsic three-dimensional character.

The occurrence of two-phase effects that depend on the flowrate immediately posesses an apparent contradiction in that, even at U=400μm/s, the system is still in the creeping flow regime (a straightforward calculation shows that the Reynolds number is well below unity at this flowrate), so that there should be no question about the assumption that the Stokes regime can be invoked. In addition, when the finite-size particle passes through the restricted gap between neighboring obstacles, it introduces a geometric perturbation to the unperturbed single phase flow. In the two-phase context, this perturbation is to be regarded as a boundary moving at a velocity $U_{\mathrm{gap}}$ that, for the geometry at hand, is about ten times larger than the average flow velocity due to the incompressible character of the flow (the restricted gas is order one tenth of the cell edge *ℓ*). The characteristic dimension, say $\ell_{c}$ over which the perturbation occurs (i.e., the distance separating peak-to-peak oscillation amplitude) is smaller than the cell edge, and this introduces an externally forced characteristic time, $\tau_{c}=\ell_{c}/U\mathrm{gap}$, in the system, for which the Strouhal number, $St=\ell /(U\;\tau_{c})$, can be significantly larger than unity (for the case in object $10^{-3}\leq \tau_{c}\leq 10^{-2}\;s$, which makes St≃2÷20. In these conditions, the product ReSt can be of order unity or above, which implies that the *unsteady Stokes equation* in place of the stationary Stokes equation should be considered to infer properties of the dynamics of the system [35]. Note that, in the system considered, this switch between regimes breaks down the pure geometric/configurational character of the steady Stokes problem, explicitly introducing a dependence of the solution on the average flow velocity *U* through the characteristic time $\tau_{c}$.

The second (three-dimensional) effect arises as a consequence of the no-slip boundary conditions due to the presence of the top and bottom walls of the channel, which induce a vertical (off-plane) velocity component above and below the symmetry plane of the structure. In [36], it is shown how, at relatively large flowrates (more precisely, at large values of the particle Peclet number), the presence of the vertical velocity profile can drastically modify the spatial distribution of the particles, causing their migration towards the floor and the ceiling of the channel. Depending on the system geometry and on the aspect ratio of the periodic cell, this effect can make the predicted value of the particle migration angle significantly different from that obtained in the two-dimensional setting of the problem.

Both effects are apt to be significant on microscale, high-throughput applications of the DLD technique, as the high flowrate data obtained in Huang’s device suggest. On the contrary, by the arguments provided above, we expect that the advection–diffusion model here proposed accounting for particle hindrance, electrostatic effects, and Brownian motion can provide a valuable tool for the design of DLD separations aimed at sorting suspended targets in the ten/hundred nanometer range (exosomes, globular proteins, viruses), which constitute the cutting edge of the current applications of this separation technology.

## 5. Conclusions

Notwithstanding the great potential exhibited by DLD-based separations for a wide variety of clinical/analytical targets, to date, practical implementations of this technique are still mostly confined to an ever growing body of tailored research prototypes, whereas a widespread standardized commercial diffusion of this type of microseparators has yet to take place. In this regard, constructing a theoretical approach—simple and reliable at one time—for predicting the dynamic features of the suspended targets may be an important cornerstone towards the development of microseparators that enforce this separation mechanism. In this article, we extended a previous transport template accounting for particle hindrance and diffusion developed by some of these authors to include possible electrostatic effects ultimately due to the existence of an electric double layer at the interface between the solid surfaces (particles, obstacles) and the suspending solution. The lumped approach proposed in [33] based on the electrostatic displacement parameter $r_{F-\mathrm{EDL}}$ to account for the effects of electrostatic interactions between particles and obstacles proved ideally suited for being included in a simple and effective way in our model accounting for particle hindrance and diffusion. Ideally, the approach proposed in this article is completely predictive in that it does not contain adjustable parameters, but just physical constants that are accessible to experimental determination. To assess its fitness to describe the real world process, we tested it against the data published in [6], specifically choosing the low flowrate experimental runs, which were those more consistent with the simplifying assumptions constituting the basis for the microdynamical model expressed by Equation (Equation 2). The comparison showed a significant improvement over previously proposed approaches. Based on the grounding assumptions of the transport template, we expect it to be particularly suited to interpret and predict the separation performance in nanoscale applications of the DLD technique.

## Figures and Tables

**Figure 1 biosensors-10-00126-f001:**
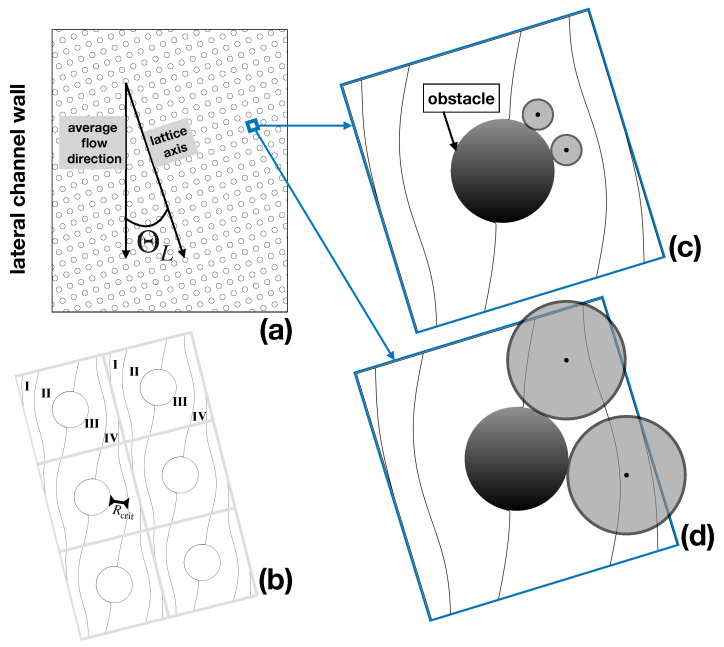
Kinematic model of particle transport. (**a**) lattice geometry; (**b**) critical streamlines defining the lanes (fluxtubes) stemming from the solution of the pressure-driven Stokes flow through the periodic lattice; (**c**) subcritical particle; (**d**) supercritical particle. The particles have been superimposed in their relative scale with respect to the unit periodic cell to the flow streamlines.

**Figure 2 biosensors-10-00126-f002:**
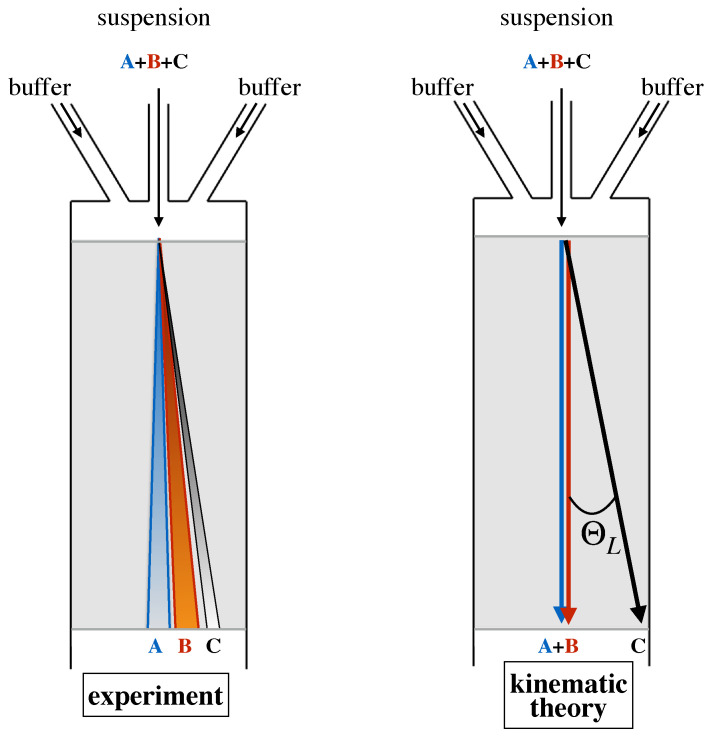
Comparison between experimentally determined behavior of particle transport in a DLD device and the prediction of the kinematic model. Particles “A” and “B” are smaller than the critical size, particle “C” is supercritical.

**Figure 3 biosensors-10-00126-f003:**
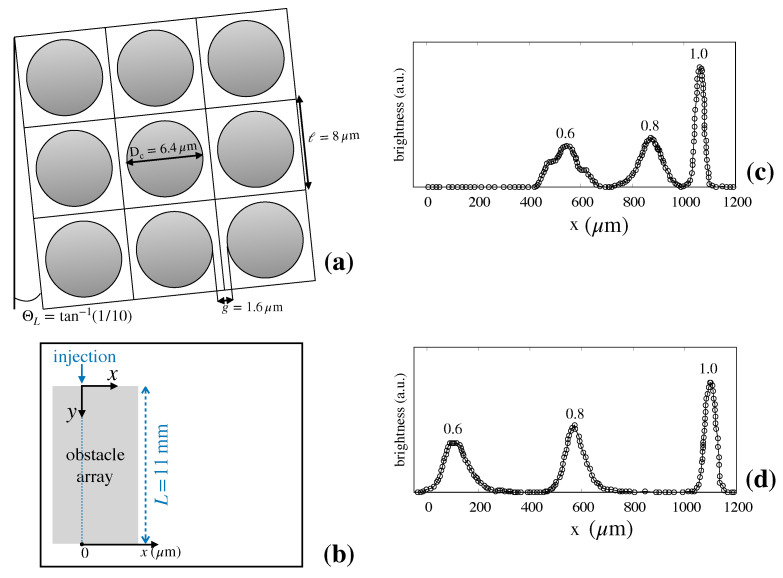
Experimental measures of particle separation efficiency in Huang’s device [6] (see the main text for details). (**a**) lattice geometry; (**b**) reference frame for fluorescence measures. The average flow direction is directed along the *y*-axis; (**c**) brightness vs. cross-section coordinate *x* at flowrate 40μm/s; (**d**) brightness vs. cross-section coordinate *x* at flowrate 400μm/s. The data reported in (**c**) and (**d**) are taken from Ref. [6].

**Figure 4 biosensors-10-00126-f004:**
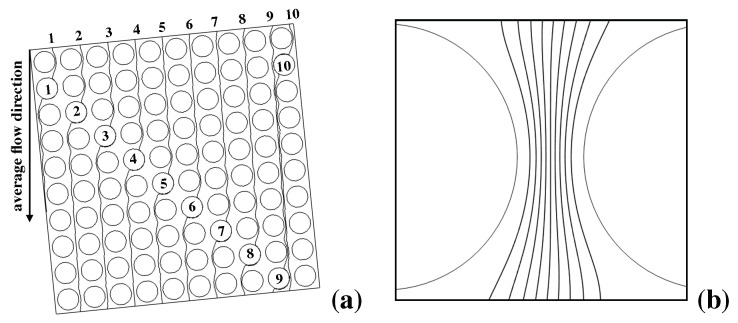
Kinematic structure of the pressure-driven Stokes flow in Huang’s prototype. (**a**) critical streamlines associated with selected obstacles (numbered from 1 to 10) in a 10×10 area of the lattice; note that each streamline enter and leaves the area at the same horizontal coordinate, its average direction being parallel to the channel walls (collinear with the average flow direction); (**b**) structure of the flux tubes (lanes) associated with critical streamlines in the restricted gap between adjacent obstacles.

**Figure 5 biosensors-10-00126-f005:**
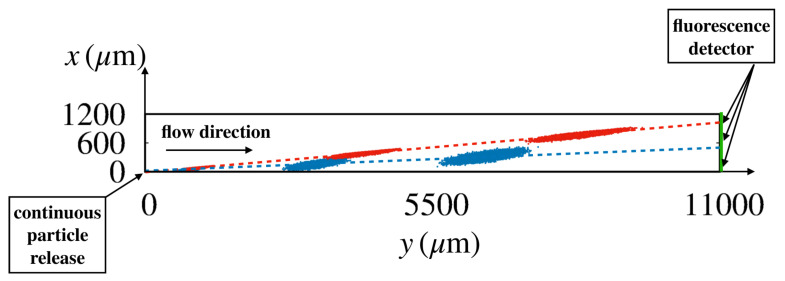
Snapshots of the instantaneous positions of the center of a particle released at the origin of the coordinate system for different realizations of the stochastic process Equation (Equation 2) in Huang’s device [6] at a flowrate equal to 40μm/s. Values of particle diffusion coefficient and flow velocity are consistent with those of the experiment (red $r_{p}=0.45\;\mu m$; blue $r_{p}=0.30\;\mu m$). The broken straight lines connect the coordinates of the center of mass of the swarm at different times. Their slope defines the average particle migration angle.

**Figure 6 biosensors-10-00126-f006:**
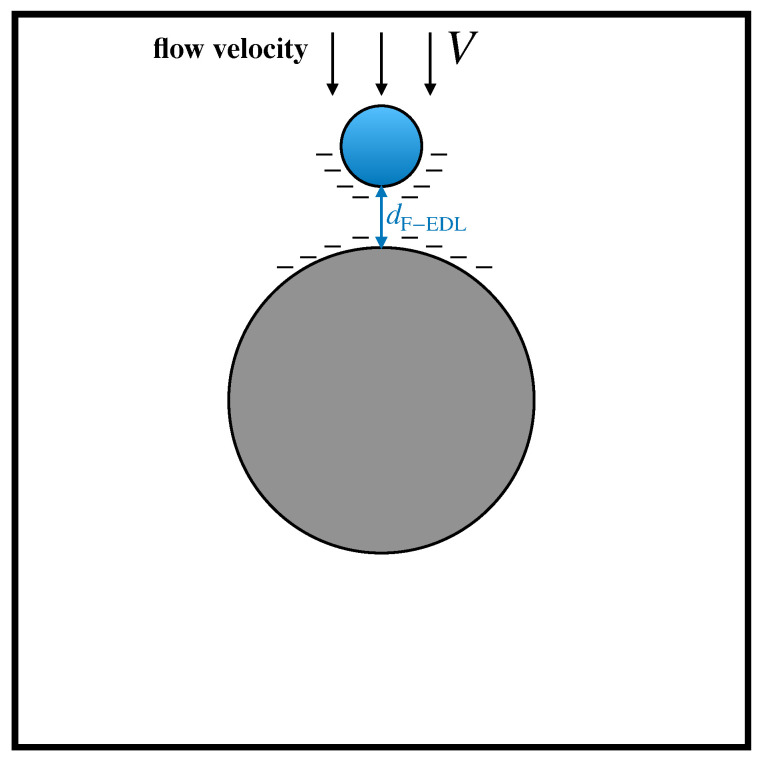
Schematic representation of the electrostatic displacement resulting from the equilibrium between electrostatic repulsive interactions and viscous drag acted upon the particle by the suspending flow.

**Figure 7 biosensors-10-00126-f007:**
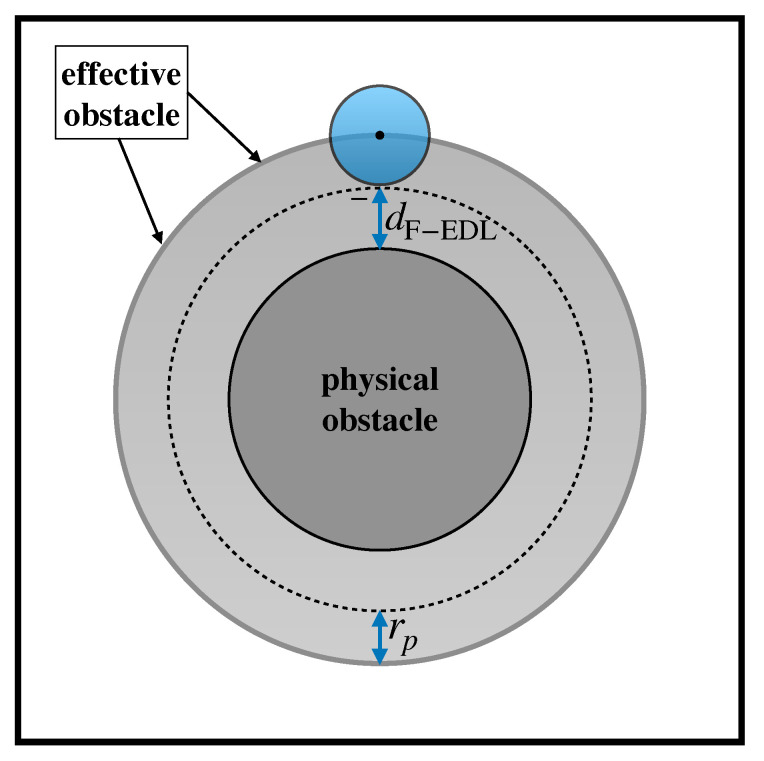
Effective obstacle approach accounting for electrostatic effects and particle hindrance. Outside the effective obstacle, a finite-sized particle is thought of as a point tracer collapsed at its center, which passively follows the streamlines of the unperturbed flow. At the effective obstacle boundary, the effect of the obstacle is such as to preserve the tangential velocity of the particle while annihilating the component normal to the boundary.

**Figure 8 biosensors-10-00126-f008:**
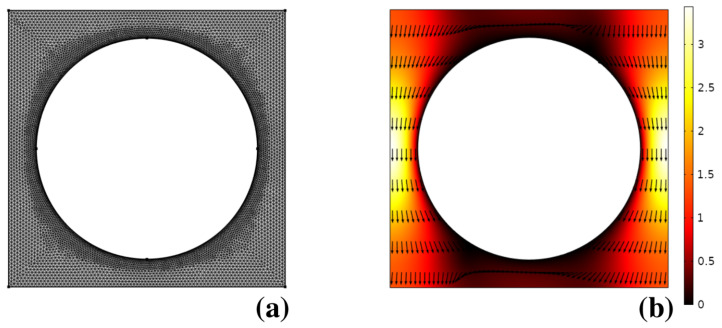
(**a**) computational mesh used for the solution of the pressure-driven laminar flow through the unit cell in Huang’s device; (**b**) flow structure. Vectors are normalized to uniform length. The contour plot depicts the velocity magnitude normalized to the average flow velocity through the unit cell.

**Figure 9 biosensors-10-00126-f009:**
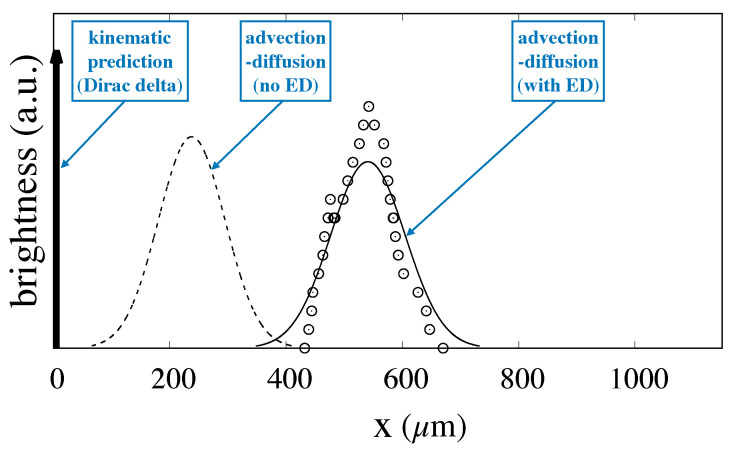
Comparison between the experimental values of the fluorescence intensity at a distance L=11mm downstream the particle feeding source for particles of size $r_{p}=0.3\;\mu m$ and the distribution resulting from the fitting of the advection–diffusion model with electrostatic effect to the experimental data. Empty symbols: experimental values taken from [6]; Continuous line: advection–diffusion model with ED computed by assuming a value of $d_{F-\mathrm{EDL}}$ that makes the mean value of the experimental and predicted distribution equal. Broken line: advection–diffusion model with no electrostatic effect. The thick arrow represents the Dirac delta distribution predicted by the kinematic theory for this system geometry and the chosen particle size. All of the distributions have been normalized so that they possess equal area.

**Figure 10 biosensors-10-00126-f010:**
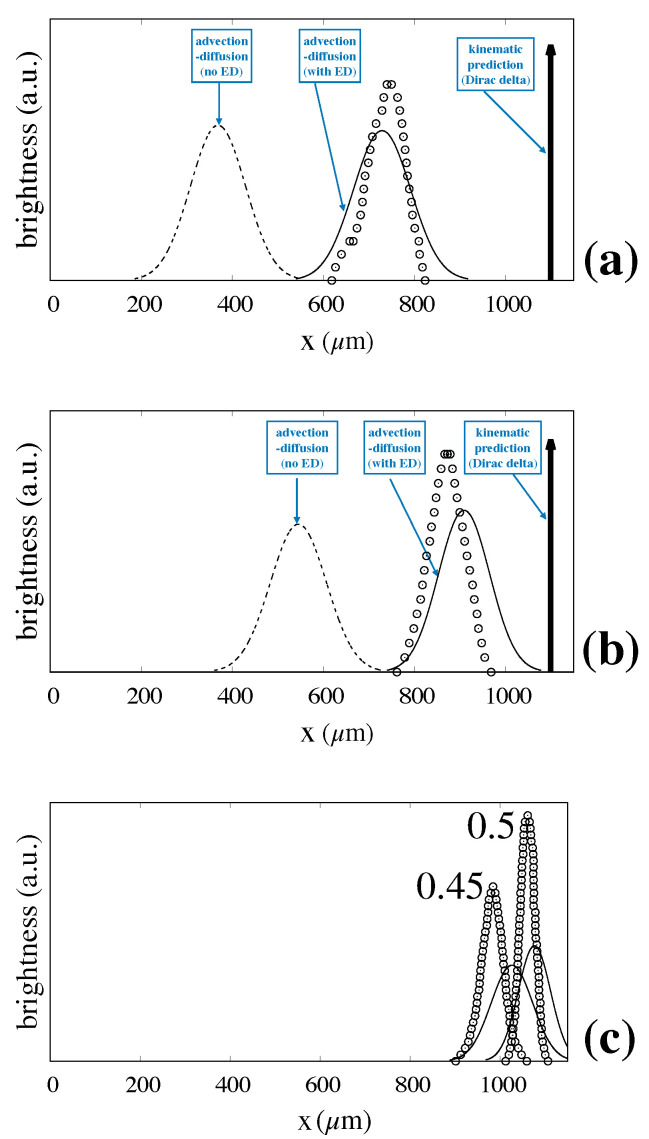
Fully predicted particle exit distributions vs. experimental values taken from Ref. [6]. Symbols and lines are consistent with those of Figure 9. (**a**) $r_{p}=0.35\;\mu m$; (**b**) $r_{p}=0.4\;\mu m$; (**c**) $r_{p}=0.45;\;0.5\;\mu m$.

**Figure 11 biosensors-10-00126-f011:**
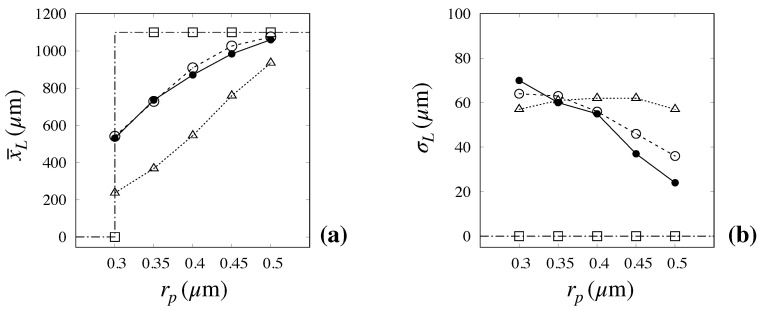
Comparison between predicted and measured values of the mean exit position, ${\bar{x}}_{l}$ (**a**), and of the distribution variance $\sigma_{L}$ (**b**) vs. particle size. The experimental values (solid bullets) are taken from [6] Dash-dotted line with empty squares: kinematic model; Dotted line with empty triangles: advection–diffusion model with no ED effect. Broken line with empty circles: advection–diffusion model with ED effect.

**Figure 12 biosensors-10-00126-f012:**
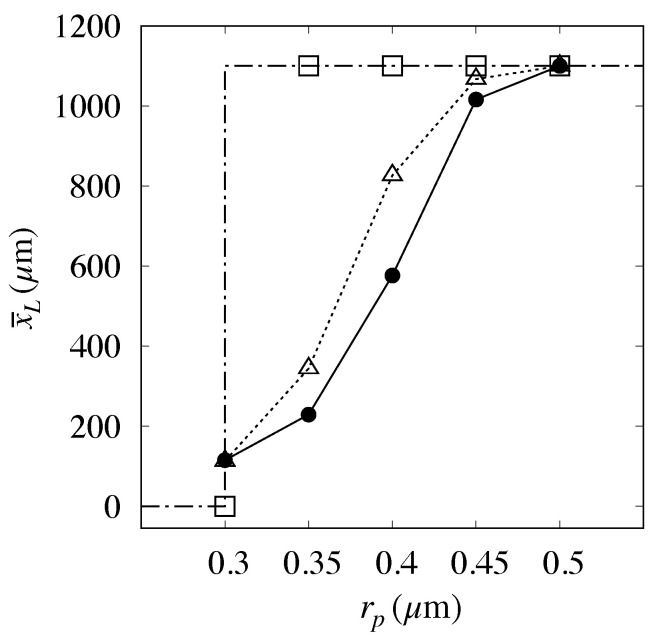
Comparison between predicted and measured values of the mean exit position, vs. particle size at a high flowrate (U=400μm/s) for Huang’s prototype [6]. Dash-dotted line with empty squares: kinematic model; Dotted line with empty triangles: Present model. Solid bullets: experimental data.

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
