# Peer review of "Combining Electrostatic, Hindrance and Diffusive Effects for Predicting Particle Transport and Separation Efficiency in Deterministic Lateral Displacement Microfluidic Devices"

_biosensors, 2020, doi:10.3390/bios10090126_

Round 1
Reviewer 1 Report
In this study, Biagioni et al. have carefully studied the effect of electrostatic effects on particle separation in DLD devices. The results presented in this work will benefit researchers in the field of nanoparticle separation, including but not limited to viruses and exosome separations. This work demonstrates that the advection-diffusion model with ED effect is better than the previous approaches. The manuscript is very well written, and I recommend it for publication.
The only minor addition I would recommend would be to simulate and experimentally validate separation performance for particles less than 100 nm diameter.
Author Response
We wish to thank the reviewer for his/her appreciation of the manuscript.
We would like to follow up the reviewer suggestion to compare experimental reults and model predictions for sub-micrometric particles, but unfortunately we have no available facilities for performing the experiments at our department. We hope that the results discussed in the manuscript can motivate further experimental investigations by other groups.
Reviewer 2 Report
Overall the paper is good - there were a few minor changes that could be addressed to improve the readability/soundness of the manuscript:
1) FIG4-B shows the fluxtubes (lanes) given for the N = 10 lattice. It may help to give a little annotation for non-specialists in DLD to understand what is being represented (especially since the lattice coordinates are rotated to be orthogonal to the viewer). Perhaps simply labeling the pillars, the flow direction, and the first fluxtube would make it more readable.
2) FIG 9 seems to suggest the performance of the model, but it is done using the 0.3µm particles which were used to tune the electro-static parameter. Although its understood the model takes into account dispersion, and therefore still shows correspondence with the experimental data, it seems incorrect to offer it as a working example of the model. FIG 10 alone seems perfectly adequate to show the predictive capabilities.
3) On line 349, the statement that the model "did not compare" at 400 µm/s seems to warrant more detail. Can the authors provide (either written or graphically) some quantification of the deviation - perhaps using the parameters introduced previously? Was it mainly in the ensemble dispersion or in the peak position, and was it uniformly inconsistent across particle size? Although its an order of magnitude higher velocity, the Re is still low and in the laminar flow regime (based on what could be ascertain from the original paper) so it would be of interest as to how much the model deviates from experiment. Additionally, although the authors invoke potential inertial effects as a reason for the deviation, given that the paper is focused on joining the previous advection-dispersion model with electrostatics, is there any reason to believe that electrostatic effects are not being taking into account at higher flow? Did the previous generations of the advection-disperison model also suffer at higher velocity in the same manner (i.e. is the electostatic parameter a non-factor)?
Author Response
We wish to thank the reviewer for his/her careful reading of the manuscript and for his/her constructive criticism, which we believe was substantial in improving the quality of the manuscript. All of the issues raised by the reviewer have been carefully considered, and changes to the manuscript have been made accordingly.
Minor typos were also corrected.
Below we report the reply to each specific point raised by the reviewer and the amendments to the manuscript.
1) Panel (a) of Fig. 4 was modified. In the current version it shows the set of critical streamlines that defines the lanes depicted in Panel (b) of the same figures. Critical streamlines have been labeled together with the associated obstacles to show the correspondence. Both the figure caption and the main text have been amended consistently with the changes in the figure.
2.) The caption of Fig. 9 was modified to explain unambiguously that the result of the advection-diffusion model with electrostatic effect show is not a prediction but a one-parameter fit to the data. Fig. 10 was also modified to avoid repetitions of the data depicted. In the current version, the cases shown in each of the figure panels are different. Thank you for pointing this out.
3.) We think that the reviewer observation is altogether correct, and that taking into account his/her observation allowed us to amend an incorrect statemement in the original version of the manuscript. In point of fact, even for the large flowrate case considered in Huang's article, a direct computation show that the Reynolds number is below unity (order 0.05), so inertial effects cannot be invoked for explaining the dependence of the average deviation angle on the flowrate, as pointed out by the reviewer. In the amended version of the manuscript, we discuss two possible mechanisms by which such dependence might arise. One is related to a three-dimensional effect that we unveiled in a previous article. The other is related to a two-phase effect. Based on an order-of-magnitude analysis of the Reynolds and Strouhal number for the case at hand, we specifically suggest that an unsteady Stokes equation might be more appropriate to describe the hydrodynamics of the system in these conditions. All in all, granted the linear character of the momentum balance equation at vanishing values of Reynolds, the time-dependent Stokes equation is the only conceptual possibility to introduce a dependence of the solution on the flowrate. Clearly, tailored experiments should be performed to either confirm or disprove this hypothesis.
This discussion is now developed in lines 362 through 394 of the amended version of the manuscript.
Again, thank you very much for your useful comments.
Reviewer 3 Report
Dear authors,
This manuscript described a very detailed theoretical model to describe the DLD+electrokinetic particle guiding or separation effect. The simulated results can match the demonstrated experimental prior arts from the references. This article was very well structured and written. Some minor suggestion are proposed here for authors' consideration.
- In this article, the simulation results were applied to the sub-micron particles. What if the proposed model is applied to the cell scale, which lies in around 10um in diameter?
- If without the electrokinetic effect, how well will this model works for sub-micron and 10's micron size particles?
- How was the Fig.1(b) generated? A detailed description should be explained on how this graph was constructed or what tool was used. This is not the same as the streamline plot.
Author Response
We thank the reviewer for his/her careful reading of the manuscript and for the comments. The points of criticism raised by the reviewer have been carefully considred, and changes to the manuscript have been taken accordingly. Please find below the reply to the specific points. Minor typos were also corrected.
1.) If the scale of the particle raises to 10 micrometers, the whole geometry and the flowrate will scale accordingly. In these conditions, we expect that the approach proposed in the manuscript may become inaccurate. We added two paragraphs at the end of Section 4 (lines 362 through 393 of the amended version of the manuscript) to point at some possible phenomena that may occur and that are not considered in our model.
2.) In the case of nanometric particles, we expect the electrostatic effect to become more and more important. Unfortunately, there are not experiments available in the literature that can be directly compared to simulations, and we hope that our results can motivate tailored experimental investigations in this range of particle size.
3.) The caption of Fig. 1 was change to better explain was is being shown.
Thank you again for your comments